# The first quantitative assessment of radiocarbon chronologies for initial pottery in Island Southeast Asia supports multi-directional Neolithic dispersal

**Ethan E. Cochrane** [1][☯]*, **Timothy M. Rieth**[2][☯], **Darby Filimoehala**[2]

**1** Anthropology, The University of Auckland, Auckland, New Zealand, **2** International Archaeological Research Institute Inc., Honolulu, Hawaii, United States of America

☯ These authors contributed equally to this work.
* e.cochrane@auckland.ac.nz

**Data Availability Statement:** All relevant data are within the paper and its Supporting Information files.

## Abstract

Neolithization, or the Holocene demographic expansion of farming populations, accounts for significant changes in human and animal biology, artifacts, languages, and cultures across the earth. For Island Southeast Asia, the orthodox Out of Taiwan hypothesis proposes that Neolithic expansion originated from Taiwan with populations moving south into Island Southeast Asia, while the Western Route Migration hypothesis suggests the earliest farming populations entered from Mainland Southeast Asia in the west. These hypotheses are also linked to competing explanations of the Austronesian expansion, one of the most significant population dispersals in the ancient world that influenced human and environmental diversity from Madagascar to Easter Island and Hawai'i to New Zealand. The fundamental archaeological test of the Out of Taiwan and Western Route Migration hypotheses is the geographic and chronological distribution of initial pottery assemblages, but these data have never been quantitatively analyzed. Using radiocarbon determinations from 20 archaeological sites, we present a Bayesian chronological analysis of initial pottery deposition in Island Southeast Asia and western Near Oceania. Both site-scale and island-scale Bayesian models were produced in Oxcal using radiocarbon determinations that are most confidently associated with selected target events. Our results indicate multi-directional Neolithic dispersal in Island Southeast Asia, with the earliest pottery contemporaneously deposited in western Borneo and the northern Philippines. This work supports emerging research that identifies separate processes of biological, linguistic, and material culture change in Island Southeast Asia.

## Introduction

The farming/language dispersal hypothesis [1] seeks to explain broad patterns of cultural, linguistic and genetic change as the result of expanding Neolithic farming populations and

**Funding:** Work by EEC was partially funded by The University of Auckland Faculty of Arts Performance Based Research Fund (no grant number or funder website). The funder had no role in study design, data collection and analysis, decision to publish, or preparation of the manuscript. There was no additional external funding received for this study. TMR and DF are employed by International Archaeological Research Institute, Inc. (IARII), a private non-profit company. IARII provided support in the form of salaries for authors TMR and DF, but did not have any additional role in the study design, data collection and analysis, decision to publish, or preparation of the manuscript. The specific roles of these authors are articulated in the 'author contributions' section.

**Competing interests:** The authors have declared that no competing interests exist. TMR and DF are employed by International Archaeological Research Institute, Inc. (IARII), a private non-profit company. This does not alter our adherence to PLOS ONE policies on sharing data and materials.

decades of research reveals that such expansions have occurred across different world regions [2–6]. In Island Southeast Asia (ISEA) (Fig 1) significant changes associated with putative Neolithic expansion include language family origins, material culture innovations, new genetic variants, increases in social complexity, and the later colonization of previously unsettled Pacific islands. The language family, Austronesian, is the pre-modern world's most geographically expansive, and probably began on Taiwan about 5000 years ago before spreading throughout ISEA, west to Madagascar, and east to the islands of Oceania [7]. Material culture innovations in ISEA, in particular the origins of pottery and ground-stone tools, are explained as the intrusive tool-kit associated with expanding farming populations [8]. The movement of these populations also generated new genetic and phenotypic variation within human, animal, and plant populations [9]. The hierarchical social system associated with later Pacific island colonization may be an innovation carried by the Neolithic groups of ISEA [10].

Considering this linguistic, biological and anthropological research, the orthodox Out of Taiwan (OoT) hypothesis formalizes Neolithic expansion in ISEA and proposes that the development of rice-based agriculture in southern China led expanding populations to disperse, with some moving to Taiwan and by approximately 4000 BP voyaging from there to the northern Philippines. In the northern Philippines, intricately decorated and red-slipped pottery was a local innovation derived from ancestral Taiwanese pottery and is "the first instance of Neolithic settlement in Island Southeast Asia outside Taiwan" [11:21]. Farming populations then spread from the northern Philippines to the south and east with pottery and other Neolithic material culture [8, 12], arriving in the Bismarck archipelago, Sulawesi, and the Mariana Islands approximately 3450–3300 BP [13:283, contra 14]. Related Neolithic populations arrived at approximately the same time in the Talauds, Sumatra, and Western Java [13:279], slightly later in the Central Moluccas by 2950 BP [13:279], and parts of Borneo by approximately 2750 BP [13:270–271].

The OoT hypothesis of Neolithic dispersal is, however, contested by archaeological, linguistic, and biological analyses. For example, after reviewing dated archaeological deposits with red-slipped pottery, Anderson [15] suggested two Neolithic dispersals. He proposed a first dispersal, Neolithic I, from southern China to Mainland Southeast Asia around 5000 BP, then to Borneo, Sulawesi, and the Moluccas via Java and Sumatra by 4000 BP (more generally referenced as the Western Route Migration [WRM] [16]). Neolithic II, a rapid, second dispersal associated with red-slipped ceramics, began from Taiwan approximately 5500 BP, to the northern Philippines, Borneo, Sulawesi, and the Moluccas all by 4000 BP. Five centuries later, this Neolithic II expansion continued to Oceania. Regarding language, Donohue and Denham argue that the previously defined hierarchical Austronesian subgroups in ISEA are not valid, nor is the geographically staged dispersal of languages those subgroups signified. Instead, current subgrouping indicates that "rapid, multi-directional, and multi-modal propagation" [17:229] of Austronesian languages better accounts for their distribution across ISEA [18]. Soares and colleagues' [19] analysis of 157 human mtDNA genomes also questions the OoT hypothesis. They found that the clade assumed to indicate Taiwanese origin of Oceanic populations originated instead in the Bismarck Archipelago several thousand years prior to Neolithic expansion in ISEA.

The OoT and WRM hypotheses, and other proposals [20, 21] are substantially based on dated archaeological assemblages, particularly those that contain ceramics [22]. Bellwood contends that Neolithic dispersal in ISEA is mirrored in "the spread of red-slipped pottery, [and] with the later additions of distinctive incised and stamped forms of decoration, appears to mark a second millennium BCE dispersal of Neolithic populations southwards from the Philippines into central and eastern Indonesia, and rapidly onwards beyond the northern coastline of New Guinea into the distant islands of Oceania" [13:302]. Carson and colleagues similarly

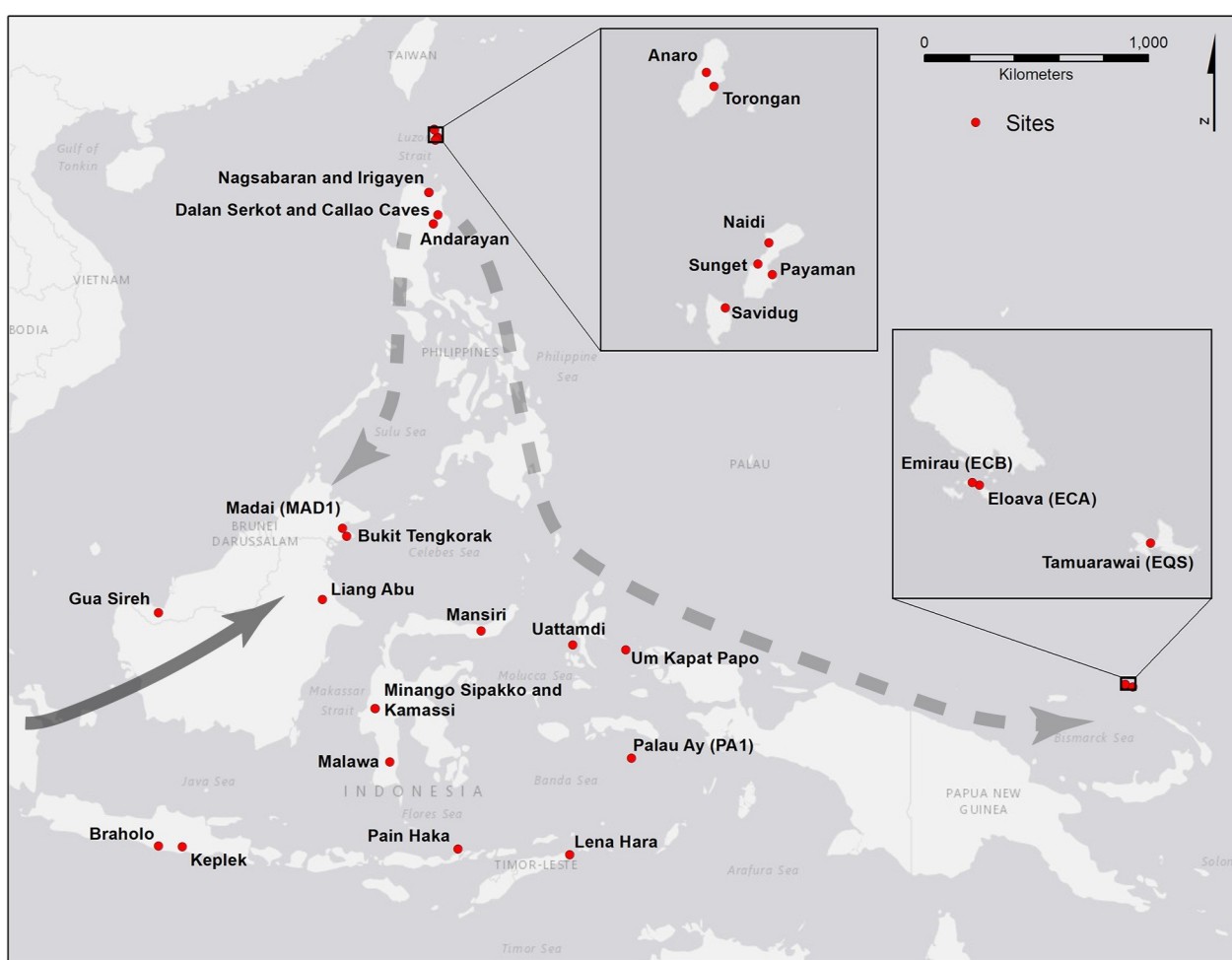

**Fig 1. Map of Island Southeast Asia and near Oceania.** Archaeological sites mentioned in the text and S1 File are shown. Arrows depict primary population movements associated with the OoT (dashed) and WRM hypotheses (solid). Basemap provided by Esri.

interpret the later decorated pottery, noting its "spread as a diagnostic tradition can be related to the spread of a cultural group" [11:17].

These and other pottery chronologies are built from individual radiocarbon determinations and while Spriggs [8, 23] has conducted chronometric hygiene analyses on individual ISEA radiocarbon determinations, the construction of chronologies from ISEA determinations (hygienically assessed or not) has never been quantitatively evaluated. Instead, to construct chronologies researchers have adopted a qualitative, and sometimes ad hoc, approach where particular determinations might be included based on a systematic hygiene protocol, but at the same time others are rejected without clear criteria (e.g., "rejected by the excavators" [8:60]). Additionally, qualitative interpretations of the length of periods in a chronology, or the onset of specific human activities [e.g., 24–26], often lack a methodological justification considering the quantitative perspective that forms the foundation of dating analyses. As Bayliss et al. state, visual inspection of dates that interprets the start, end, or duration of activity "has no mechanism to allow for the statistical scatter on the radiocarbon dates. . .[and] [i]n effect, the scatter and uncertainties on the radiocarbon dates are being confused with longevity of ancient activity" [27:9]. The qualitative procedures of chronology construction, even with the use of chronometric hygiene protocols, may result in the favoring of preferred dispersal hypotheses [e.g., 28, 29].

To avoid this we present a model-based Bayesian chronological analysis of dated mid-Holocene pottery assemblages in ISEA and Mussau, Bismarck Archipelago, to determine if directional dispersal of pottery assemblages can be identified. This is largely the same dataset that others have used to develop extant regional dispersal chronologies. The model-based Bayesian approach to calibration and the construction of chronologies contrasts with visual inspection of individually calibrated dates, which assumes that a particular date or suite of dates is equivalent to an archaeological event of interest (a conflation of the radiocarbon and target events [30]). Setting aside depositional processes that potentially confound the association of radiocarbon dates and archaeological events, visual inspection of dates is also fallible in its assumption that the dates are a representative sample of the population of dates that could be obtained from a particular context.

The results of our Bayesian chronological analyses of 20 archaeological sites across six island regions do not conform to the expectations of the orthodox OoT or the Western Route Migration hypotheses and instead indicate contemporaneous pottery deposition in Borneo and the northern Philippines, with the geographic pattern of subsequent ceramic deposits varying depending on site- or island-scale analyses. We conclude that quantitative analysis of the valid suite of earliest pottery radiocarbon determinations cannot differentiate uni-directional dispersal hypotheses and instead currently supports multi-directional Neolithic dispersal in ISEA. This requires a revision of our understanding of the Neolithic in ISEA, the ancestry of Oceanic populations, and the archaeological, linguistic and genetic research that underlie them.

## Results

Our results are presented at site- and island-scales depending on the distribution of radiocarbon determinations. All referenced models and radiocarbon determinations are in the S1 File and a summary of the model results is given in Table 1. We discuss the 95.4% highest posterior density (HPD) results, but also include the 68.2% HPD results in Table 1. All of the models exhibit stability in results over the course of five runs: convergence values remain >95% and the date ranges for the start dates remain consistent.

### Batanes Islands

We calculated the initial appearance of pottery on Batan Island with two models (S1 Fig in S1 File). Based on a model that only includes determinations from unidentified charcoal this event occurred sometime between *3380–2590 cal BP (95.4%)*. When three determinations obtained from residue on sherds are included, the date is *3640–2890 cal BP (95.4%)*. The residue-derived determinations add about 250 years to the date (shifting the entire date range, though it still overlaps with the first model's date), though the reason for this apparent inbuilt age is unclear (reported δ¹³C values are consistent with fully terrestrial samples as opposed to marine taxa). Both island-scale models provide younger and more precise dates than those from the single site on the island with a sufficient number of radiocarbon determinations for site-scale modelling, Sunget, where initial appearance of pottery occurred between *4330–2890 cal BP (95.4%)*. This is due to the increased number of determinations included in the single-phase island model. The dated Batan Island pottery assemblages include red-slip and circle-stamp surface treatments.

Itabayat Island pottery has older dates than Batan. Based on radiocarbon determinations from the Torongan site and different areas of the Anaro Hilltop site, pottery entered the archaeological record between *5200–3990 cal BP (95.4%)*. The radiocarbon determinations from the Torongan site, also on residue, influence the older range of this date. Red-slipped pottery is associated with these determinations at the base of the cultural layer. The Anaro Hilltop model with nine

**Table 1. Modeled dates.**

| Island/Region | Site[a] | Ceramic characteristics | No. of [14]C Determinations per Model[b] | Start, cal BP (95.4%) | Start, cal BP (68.2%) |
|---|---|---|---|---|---|
| **Batan** | | | 12 | 3640–2890 | 3310–2990 |
| **Batan (no residue determinations)** | | | 9 | 3380–2590 | 2970–2730 |
| | Sunget | red-slip, circle-stamp | 5 | 4330–2890 | 3490–3000 |
| **Itabayat** | | | 11 | 5200–3990 | 4620–4150 |
| | Anaro Hilltop | circle-stamp | 9 | 3470–2760 | 3070–2790 |
| **Luzon**[c] | | | 24 | 5430–4290 | 5280–4370 |
| | Nagsabaran | red-slip, circle-stamp, impressed, dentate | 27 | 5440–4270 | 5100–4390 |
| **North Borneo** | | | 5 | 4560–2460 | 3360–2730 |
| **Borneo (all)** | | | 7 | 6680–3820 | 5230–4170 |
| | MAD1 | red-slip, impressed | 5 | 8580–2710 | 8530–2730 |
| **Sulawesi** | | | 9 | 4550–3590 | 4180–3740 |
| | Minango Sipakko | red-slip, impressed, incised | 6 | 4600–3510 | 4120–3640 |
| **Flores** | Pain Haka | red-slip, incised, applique | 10 | 3200–2600 | 3010–2750 |
| **Pulau Ay** | PA1 | red-slip, incised | 7 | 3740–3020 | 3510–3210 |
| **Eloaua** | | | 23 | 3470–3010 | 3350–3070 |
| | ECA | red-slip, incised, circle-stamp, dentate | 20 | 3460–3010 | 3330–3080 |

[a]Some islands are characterized by single sites. In these cases the island date range and site date range are the same.

[b]Earliest ceramic deposition calculated for islands and sites (HPD for start boundaries of basal ceramic-bearing deposits). For multi-phase models, the number of radiocarbon determinations includes those from pre- and/or post-ceramic phases.

[c]The Luzon Island model comprises fewer determinations than the Nagsabaran site model as the 16 dates (out of 33) from the preferred Nagsabaran model were combined with 8 dates from other sites.

determinations dates the appearance of pottery there between *3470–2760 cal BP (95.4%)*. The pottery associated with the Anaro Hilltop date includes circle-stamp surface treatment.

## Philippines

Pottery is calculated to have entered the archaeological record of Luzon Island, northern Philippines, between *5430–4290 cal BP (95.4%)* (S2 Fig in S1 File). This date derives from 24 radiocarbon determinations from five sites: Andarayan, Irigayen, Nagsabaran, Dalan Serkot, and Callo Cave. Only Nagsabaran has a sufficient number of determinations for site-scale modelling. Red-slip, punctate, incised, impressed and black-incised pottery is associated with the radiocarbon samples at Andarayan, Irigayen, Dalan Serkot and Callo caves.

The Nagsabaran shell midden site, across the river from Irigayen, has seen numerous excavation campaigns and a variety of materials from the site have been analyzed including ceramics and faunal remains. Nagsabaran contains a diverse early pottery assemblage in the lower silt layer with red-slip, impressed, circle-stamp, and dentate surface treatments. This stratum is capped by a shell midden layer. Because there is more published information for Nagsabaran compared to most Neolithic pottery sites in ISEA, we were able to construct eight models to evaluate the effects of systematically including or removing determinations: (1) a single-phase model including all determinations for the lower silt layer; (2) a single-phase model for the lower silt layer excluding the oldest determination; (3) a single-phase model including all determinations from the upper shell midden layer; (4) a single-phase model for the upper shell

midden layer excluding the oldest determination; (5) a two-phase model including all determinations; (6) a two-phase model excluding the oldest determination from both the lower silt layer and the upper shell midden layer; (7) a two-phase model excluding the oldest silt layer determination and the five oldest shell midden determinations, all on a single shell species; and (8) a two-phase model using those determinations preferred by Hung et al. [31]. All model results and arguments for including and excluding particular radiocarbon determinations are in the S1 File. Here we present the model (7) results as this model has the greatest sample size, excludes the anomalous lower silt layer determination, and avoids the possibly confounding issue of whether a localized correction value is needed to calibrate the freshwater shell [32]. The larger sample size should improve the accuracy of the results. With this model, the beginning of ceramic deposition was sometime during *5440–4270 cal BP (95.4%)*.

## Greater Sunda Islands

Six single- or multi-phase models were created for site deposits on Borneo and Sulawesi Islands. The radiocarbon determinations from these sites were combined with individual determinations at other sites or excavation areas to create island-scale models and, in the case of Borneo, a sub-island model (S3 Fig in S1 File).

The date for the initial appearance of pottery at the MAD1 site on Borneo is between *8580– 2710 cal BP (95.4%)*. This date is based on a preferred non-shell model that produces negligibly different results from the model including a shell determination (both models are in the SI). The associated deposit has red-slip and impressed pottery.

The charcoal and rice husk radiocarbon determinations from the basal ceramic deposits at MAD1, Liang Abu, Bukit Tengkorak, and Gua Sireh were combined to produce a Borneo-wide model which dates the earliest appearance of ceramics at *6680–3820 cal BP (95.4%)*. Due to the size of Borneo and the possibility of spatial variation relevant to distinguishing dispersal hypotheses, a northern Borneo model using Liang Abu, Bukit Tengorak, and MAD1 determinations was produced and this dates the earliest appearance of ceramics to be sometime between *4560–2460 cal BP (95.4%)*.

The site of Minango Sipakko on Sulawesi Island has multiple excavation units and the model dating the initial appearance of ceramics includes determinations from four of these. The site-scale model date is *4600–3510 cal BP (95.4%)* and is associated with red-slip, impressed, and incised ceramics. The determinations from Minango Sipakko were combined with two determinations from Malawa and a single determination from Mansiri to produce an island-wide date for the earliest appearance of ceramics on Sulawesi sometime during *4550– 3590 cal BP (95.4%)*. The early Sulawesi ceramic deposits contain red-slip, impressed, incised, and circle-stamped sherds.

## Lesser Sunda Islands

There is a single site in the Lesser Sundas with enough radiocarbon determinations to model initial pottery deposition. The initial appearance of ceramics at the Pain Haka site on Flores Island occurred sometime between *3200–2600 cal BP (95.4%)* based on a model that applies mixed atmospheric-marine curves to calibrate dates from human bone (S4 Fig in S1 File). The Pain Haka ceramics are from burials and include red-slip, incised, and applique surface treatments. The Pain Haka site model is the same as the Flores Island model.

## Molucca Islands

One site provides information on the appearance of pottery in the Moluccas (S3 Fig in S1 File). The initial appearance of ceramics at PA1 on Pulau Ay occurred sometime between *3740–3020*

*cal BP (95.4%)*. The assemblage contains red-slip and incised sherds, with circle-stamp appearing in more recent deposits. The PA1 model is also the island model for Pulau Ay.

## Mussau Islands

Two single-phase models were created for site deposits on Eloaua Island (S4 Fig in S1 File). Twenty determinations from site ECA on Eloaua provide a date for the initial appearance of ceramics sometime between *3460–3010 cal BP (95.4%)* An island-scale model that combines the ECA determinations and three determinations from site ECB provides a date for the appearance of ceramics on Eloaua sometime during *3470–3010 cal BP (95.4%)*.

## Discussion

### Qualitative and Bayesian analysis comparisons

Summary chronologies based on qualitative analyses have been proposed for many sites and island-regions in ISEA. Comparing these chronologies with the Bayesian results generated from largely the same datasets suggests some of the qualitative analyses variably suffer from bias towards favored hypotheses, spurious precision, and few clear evaluative criteria.

All of our dates for the appearance of pottery on specific islands and sites of the Batanes Islands (e.g., Batan Island, *3380–2590 cal BP*, and Anaro Hilltop Site, Itabayat Island, *3470–2760 cal BP*) encompass Anderson's [15] proposed date of around 3000 BP. In contrast, Bellwood and Dizon [33] argue that earliest pottery in the Batanes Islands appears about 4500 BP at the Torongan site on Itabayat. This date is associated with a radiocarbon determination on residue and seems to come from the oldest end of the calibrated range of the single date (OZH-711), despite a younger date (Wk-14642), also on residue, from the same depth and context. As we have noted, and echoing Anderson [15], radiocarbon determinations on residue from sherds are older than determinations on charcoal from the same contexts. For example, at the Savidug Dune site (not included in our results, but see S1 File), the conventional radiocarbon age (CRA) on residue (Wk-21810) is more than 700 years older than that on charcoal (Wk-21808) from the same deposit, while at the Sunget Main Terrace site, the two residue determinations (Wk-14640 and ANU-11817) are a little over 500 years older than the charcoal CRA (Wk-15649) from the same deposit. Bellwood and Dizon also use radiocarbon calibrations on shell from the Torongan Site with a ΔR value derived from the Paracel Islands in the middle of the South China Sea, not a locally derived value [33: Table 5.1] and this likely imparts additional unrecognized dating error.

An island-scale Bayesian calibration model for Batan Island that excludes the residue determinations provides a younger and less precise date at 95.4% HPD, *3380–2590 cal BP*, but one similar to the only other site-scale, non-residue model, that for the Anaro Hilltop site on Itabayat Island at *3470–2760 cal BP* (95.4% HPD). This raises the possibility that "residue" is a problematical dating material, as Bayliss and Marshall [34] have shown for England and residue dates may be too old, relative to archaeological target events. Overall, Bellwood and Dizon's [33] chronology for the Batanes Islands appears biased towards a favored dispersal hypothesis as it uses questionable dating material, both residue and shell without a verified correction factor, and arbitrarily rejects or accepts particular samples and date ranges.

On Luzon Island, Philippines, the Nagsabaran site 95.4% probability for the earliest appearance of ceramics *(5430–4290 cal BP)* is earlier and less precise than published age ranges for earliest ceramics in the local area (Cagayan Valley): Hung et al.'s [35] 3950–3250 BP range and Carson and Hung's [29] 4150–3950 BP range, published more recently. Unfortunately, this important site is dogged by chronological issues: date-depth inversions, contradictory descriptions of site formation, lack of isotopic or identification data for dating samples, and ad hoc

dating analyses [see S1 File and 36: footnote 2]. The Nagsabaran 95.4% probability is based on the largest suite of potentially valid determinations, yet is still a wider date range than that proposed by previous analyses. Moreover, the best dated site in our analyses—ECA on Eloaua Island, with 20 samples, the majority of these identified to taxon, and relevant provenience information—has a modelled range of 450 years (95.4% HPD) for the onset of ceramic deposition. This, and our analysis of the Nagsabaran dating samples, suggests that the precision of Carson and Hung's analyses (e.g., 200 years), and similarly precise dates across ISEA, is spurious.

Although this might also be said of more areas in ISEA, in the Greater Sunda Islands the widely varying CRAs and error ranges of individual determinations hamper the building of precise Bayesian dates and identification of any consistent relationship between these dates for the earliest pottery and chronologies derived from qualitative analyses. Simanjuntak [16: 203, 207], for example, proposes that the earliest pottery-using populations in Sulawesi may date to 4000 BP, within our date range for the island (*4550–3590 cal BP*), and that northern Borneo was settled by pottery-using groups from the northern Philippines a bit earlier, around 4500–4000 BP [16: 202, Figure 11.1], the opposite of our chronological order analysis (see next section). Anderson's [15: Figure 5] graphic depiction of Neolithic I and II indicates the arrival of pottery-using populations in Borneo and Sulawesi between 4500–4000 BP, encompassed by the Borneo models with their large ranges (northern Borneo: *4560–2460 cal BP*; Borneo: *6680–3820 cal BP*) and the Sulawesi model. Finally, Bellwood [37:194] suggests a Neolithic expansion from the Philippines into northern (Malaysian) Borneo and Sulawesi beginning approximately 3450 BP. This fits within our northern Borneo island model, but is about a century more recent than the oldest range included in the Sulawesi model.

While the qualitative dates of initial pottery deposition for islands and regions in ISEA show little consistent relationship with our Bayesian models, in the Mussau islands previous dating proposals are generally consistent with our modeled date ranges for the first appearance of pottery. Kirch [38], for example, proposes that site ECA was occupied at least by 3300 cal BP and perhaps 150 years earlier, and both dates are encompassed by the 95.4% modeled date range (*3460–3010* cal BP). It seems most likely that the agreement between qualitative chronologies and our dates is a product of the current focus on identified dating samples, valid correction factors, and greater attention to consistent evaluative criteria [39, 40] in Mussau compared to many areas of ISEA. Nagsabaran, with 33 published dates, suggests that the agreement between Kirch's Mussau analyses and ours is not simply a function of the relatively large number of dated samples.

## Comparing Neolithic dispersal hypotheses

In most instances, the dates for first appearance of pottery in ISEA and Near Oceania return large date ranges (see Table 1). This is a product of few radiocarbon determinations and simple model structures (due to a limited number of sites with multiple dated strata). Importantly, and despite the large date ranges, because our chronological calculations are generated with quantified uncertainty, we can use computational techniques to compare dates and assess current dispersal hypotheses in ISEA. Figs 2 and 3 illustrate the chronological ordering of the appearance of pottery at site- and island-scales, respectively (S1 and S2 Tables in S1 File).

The chronological ordering of dates for earliest pottery deposition can be compared to the contrasting expectations of Neolithic dispersal hypotheses for ISEA. The OoT hypothesis proposes that pottery-using populations first entered ISEA from the north arriving on Luzon Island, northern Philippines, around 4000 cal BP. From there, groups moved southeast arriving in Sulawesi and the Bismarck Archipelago around 3450–3300 cal BP, at about the same

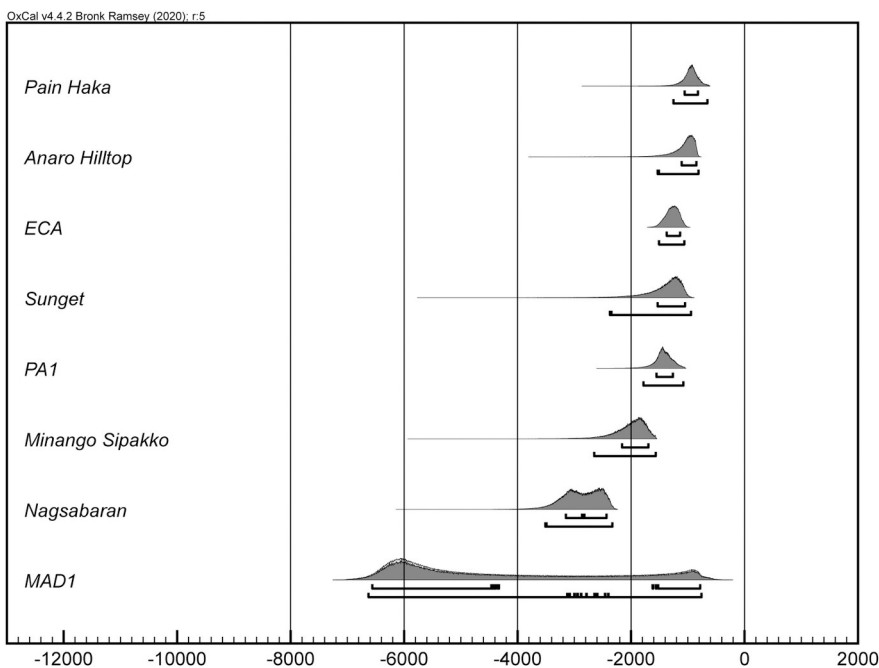

**Fig 2. Order of initial pottery appearance at archaeological sites.** Chronological ordering of the HPDs for the start of ceramic deposition at a site-scale. The first lines beneath each distribution delimit 68.2% HPD and the second lines delimit 95.4% HPD.

time that related populations arrived in parts of the Lesser Sundas, Sumatra and Java, preceding the arrival of Neolithic groups in the Moluccas and areas of Borneo [11, 13, 29]. The Neolithic I/II [15] or Western Route Migration (WRM) [16] hypotheses suggest that earliest ceramic deposition in Borneo approximately 4500 cal BP, precedes Luzon. This represents the earliest ISEA Neolithic population, Neolithic I, and is distinct from the population that settled the northern Philippines, labeled Neolithic II. Both Neolithic streams, however, converged in the Greater and Lesser Sundas and Moluccas.

The chronological ordering of dates for first ceramic deposition does not clearly support either the WRM or OoT hypotheses. At *island-scale*, the first ceramic deposits are found on Luzon, then Borneo, Itabayat (Batanes Islands), then Sulawesi, while at *site-scale*, ceramic deposits are found at MAD1 (northern Borneo), then Nagsabaran (Luzon), and then Minango Sipakko (Sulawesi). As the chronological order of the Borneo and Northern Philippines ceramics depends on how we aggregate radiocarbon determinations (i.e., at site- or island-scale), the conservative interpretation of the model results is of contemporaneous deposition of ceramics in these areas, at least within radiocarbon dating precision. After this, ceramics appear in the Batanes Islands, Sulawesi, and the Moluccas in different possible orders depending on site- or island-scale analyses. The earliest Mussau and Lesser Sunda Islands ceramics are deposited near the end of the chronological order. The lack of consistent directionality across the site- and island-scale chronological orders, from west to east, or north to south supports proposals of multi-directional movement of Neolithic populations and ideas across ISEA [20, 41, 42], after the first Neolithic deposits in the northern Philippines and Borneo.

More specific dispersal hypotheses might also be assessed with the modeled date ranges. For example, Carson and colleagues [11] argue that the dentate, incised, and red-slip ceramic assemblage at the Nagsabaran site was deposited before similar Lapita assemblages in the Mussau Islands. Our chronological ordering of dates (Fig 2 and S2 Table in S1 File) indicates that

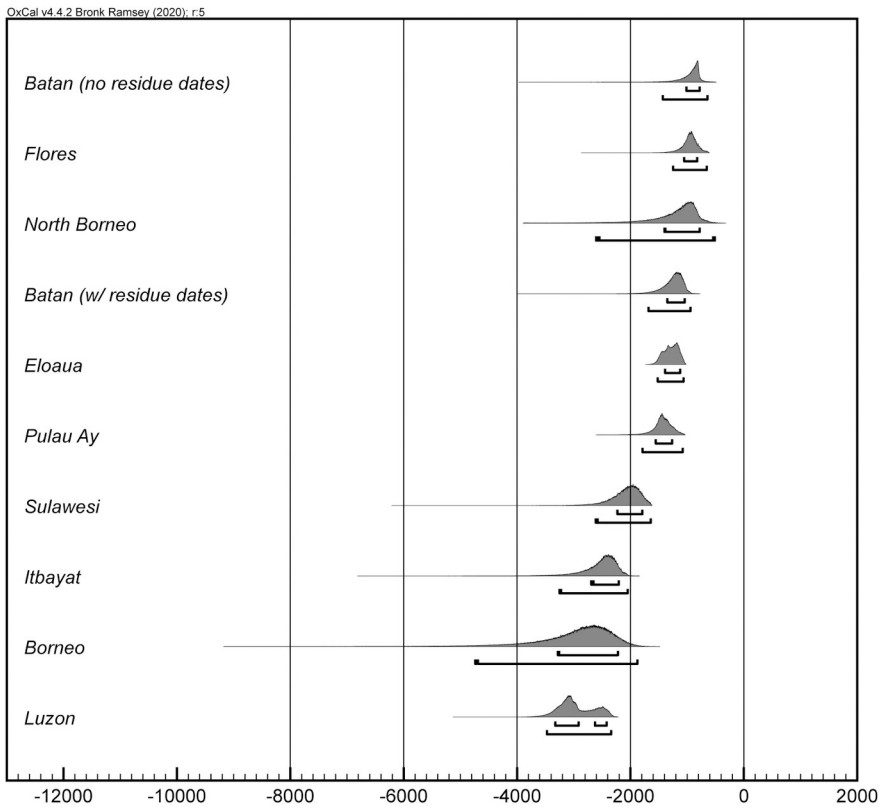

OxCal v4.4.2 Bronk Ramsey (2020); r:5

Batan (no residue dates)

Flores

North Borneo

Batan (w/ residue dates)

Eloaua

Pulau Ay

Sulawesi

Itbayat

Borneo

Luzon

-12000  -10000  -8000  -6000  -4000  -2000  0  2000

**Fig 3. Order of initial pottery appearance on islands.** Chronological ordering of the HPDs for the start of ceramic deposition at an island-scale. The first lines beneath each distribution delimit 68.2% HPD and the second lines delimit 95.4% HPD.

this site-scale relative chronology is correct. Carson and colleagues further suggest that the relative chronology, ceramic surface-treatment similarities, and linguistic analyses indicate that a population from Luzon migrated to the Bismarck Archipelago where they or their direct descendants produced Lapita ceramics. However, the chronological ordering of site-scale dates suggests other possibilities: pottery at MAD1 is chronologically earlier than both Nagsabaran and the ECA (Mussau Island) dates (Fig 2 and S2 Table in S1 File) and the earlier MAD1 assemblage contains surface treatments—red-slip and impressed (or punctate)—found at the later Nagsabaran and ECA sites (and other Mussau sites). Thus, both the Nagsabaran and ECA ceramic assemblages may share derived homologous similarity [sensu 43] from Borneo, while post-ECA assemblages such as at Pain Haka on Flores, and possibly Mansiri on Sulawesi (see S1 File), may share ancestral or derived homologous similarity from Borneo, Mussau (or Near Oceania generally) and the northern Philippines. These proposed multidirectional phylogenetic homologous relationships between ceramic assemblages across ISEA and Near Oceania, are empirically testable with appropriate ceramic classifications [e.g., 44] and are supported by research on several dimensions of human variation. Linguistic [45, 46] analyses, for example, demonstrate that some Austronesian cognates and morphosyntactic structures are likely a product of the spatial proximity of language communities and a general east-west movement of people and ideas across ISEA and Near Oceania. Concomitantly, the lack of deep structure in ISEA Austronesian languages, as well as the lack of ancestor-descendant ordering of Austronesian languages in ISEA [47] (i.e., Malayo-Polynesian languages), can suggest, instead of a single expansion pulse [cf., 7, 11], multiple population movements and

linguistic transmission over the millennium or more of initial ceramic deposition in the region [41, cf. 48]. Human and commensal biology can also be readily interpreted in light of frequent multi-directional movements. Lipson et al.'s aDNA analysis [49], for example, indicates multiple human movements from the north and west, although they favor the OoT hypothesis as a "primary movement" [49:5]. A later analysis of Y-chromosome and mtDNA by Soares et al. [50] is congruent with our results and concludes that instead of migration pulses or significant population movements, small-scale events, "a minor Neolithic input from MSEA [Mainland Southeast Asia], probably immediately preceding a Neolithic input from Taiwan", account for contemporary genetic patterns. Genetic analyses of commensal animal species also suggest multi-directional movement across ISEA [51, 52], while those of paper mulberry [53] and rice [54] document movements southward from Taiwan and eastward from Mainland to Island Southeast Asia, respectively. Significantly, the recent analysis of rice genomes and ecologies indicates that in ISEA, including the Philippines, rice derives from MSEA, contradicting previous arguments [12] for a rice-agriculture fueled expansion from Taiwan south into ISEA. Finally, multi-directional movement during the period of initial ceramic deposition is also evidenced from analyses of obsidian artifacts and domesticated plant microfossils [55, 56].

## Conclusion

Our results are the most accurate chronological calculations based on currently available data. The general lack of precision in the Bayesian model-derived dates for the initial deposition of ISEA ceramic assemblages suggests that the precision in many qualitative date estimates is likely spurious. This is particularly true for deposits with disturbed stratigraphy, a common occurrence that Swete Kelley [22:404] notes "has brought the contextual reliability and the selective use of dates within the associated radiocarbon chronology into question" across ISEA.

Considering the prominent OoT and WRM hypotheses for Neolithic dispersal in ISEA, our Bayesian calibration-derived dates support contemporaneous appearance of pottery in Borneo and the northern Philippines and do not reveal singular or primary routes of dispersal in the region. Importantly, our work highlights the current poor quality of the archaeological radiocarbon record in ISEA. Assessment of detailed dispersal hypotheses is hampered by this record. More radiocarbon determinations on suitable sample-types, with both clear links to archaeological events and valid correction factors, are required. And as we generate more accurate and precise dates we can begin to explore the implications of possible multi-directional Neolithic dispersals in ISEA [cf. 57]. This should include moving beyond the categorical framing of human history in ISEA and Oceania (e.g., fast-train versus slow-boat [58]; two-layer versus local evolution [59]; Austronesian versus Papuan [60]) to address complex processes that shaped continuous variation in past populations.

## Materials and methods

No site deposit included in our analysis is considered to have a sufficient number or quality of radiocarbon determinations to provide highly precise dates for the onset of ceramics, though ECA on Eloaua Island and Pain Haka on Flores have relatively robust radiocarbon suites. Since models with a small number of radiocarbon determinations result in imprecise dates (and due to other factors [e.g., inbuilt age for unidentified charcoal], possibly inaccurate results), only the model results for deposits or islands with five or more radiocarbon determinations are considered in our analysis; models for deposits or islands with less than five radiocarbon determinations are, however, included in the S1 File for comparison. We present results at 95.4% and 68.2% highest posterior density (HPD) intervals. Our interpretations rely on the 95.4% HPDs as they provide greater accuracy.

Radiocarbon determinations were collated from an extensive search of the ISEA and Mussau (Near Oceania) archaeological literature (S3 Table in S1 File). We did not analyze radiocarbon determinations from mainland Southeast Asia, southern coastal China, or Taiwan as the OoT and WRM hypotheses, and other proposals, are based primarily on ISEA archaeological deposits. Dating samples include unidentified and identified charcoal, uncharred wood, charred residue on pottery, animal and human bone, and freshwater, estuarine, and marine shells. We did not apply strict chronometric hygiene [8, 23] or date classification [61] protocols as either would greatly diminish sample size; however, we do evaluate the effects of particular dating materials on dating results (see S1 File). The following criteria were applied to selecting radiocarbon determinations for analysis:

1. The radiocarbon determination was associated with a distinct recovery unit (e.g., an excavation pit) that sampled a deposit including the earliest appearance of putative Neolithic ceramics at a site; determinations with ambiguous stratigraphic associations were excluded from analysis, but they are included in S3 Table in S1 File. We do not analyze determinations associated with Chinese porcelains or ceramics of well-documented Metal Age time frames.

2. A deposit comprises a single depositional event and may be a stratum, lens, feature, or individual artifact (e.g. a sherd with charcoal residue). All radiocarbon determinations from a deposit with Neolithic ceramics were included in S3 Table in S1 File as we are interested in the population of radiocarbon determinations associated with an event, not simply those with the oldest CRA; that is, the determinations from a deposit are considered to be a sample of the population of determinations. We only analyze determinations that are reasonably associated with the deposit of interest and we disregard determinations for which a clear depositional-association cannot be made (e.g., due to unmistakable stratigraphic mixing), although we do provide these determinations in S3 Table.

3. For recovery units with dated stratified deposits, determinations from stratigraphically inferior aceramic deposits (pre-dating the basal Neolithic ceramic-bearing stratum) or superior deposits (post-dating the basalt Neolithic ceramic-bearing stratum) were included. These older and younger radiocarbon ages were incorporated as *terminus post quem* and *terminus ante quem* constraints in our modeling, respectively.

4. Dating materials were evaluated for calibration issues. These include potential inbuilt age (unidentified charcoal), the lack of localized correction values (marine shell and other marine-influenced samples, possibly freshwater shell), and the lack of isotopic data (e.g., for animal bone) for determining the use of a mixed atmospheric-marine calibration. Modeling parameters to address these factors of uncertainty are discussed below.

Additionally, as there are different pottery surface-treatments in the region that potentially have chronological and culture-group links [22], we described the pottery associated with particular radiocarbon determinations using the presence-absence of general surface treatments as identified by other researchers (see S1 File). These categories include red-slip, orange-slip, circle-stamped, incised, impressed, dentate, black, and cord-marked. We included presence-absence data because abundance measures are inconsistently reported; nor are abundance measures necessary for our analyses. We did not divide these ceramic categories more finely as some have (e.g. different kinds of circle-stamping). Like others, we assume that these categories track homologous relationships amongst pottery-using groups, but in the future a problem-oriented classification is needed to define homologous links [e.g., 62].

Oxcal 4.3.2 [63] was used for Bayesian model calibrations and the generation of datesfor the initial appearance of ceramics in the archaeological record at site- and island-scales. To re-

state Bayes' rule, the models structure radiocarbon determinations (standardized likelihoods in Bayesian statistics) by the relationships between contexts (prior beliefs) to provide the dates (posterior beliefs). Further, these models assume that the radiocarbon determinations are uniformly distributed (uninformative prior belief). Oxcal calculates the probability distributions of the individual dates and uses Markov Chain Monte Carlo (MCMC) sampling to calculate the best possible posterior values given the data and model structure. Thus, events that are not directly datable—the start and end of deposition, for example—and the duration of events can be quantified with statistical confidence. For this study, the date ranges of interest are the start for the oldest ceramic-bearing deposits for site-scale models and the start for the composite single ceramic phases for the island-scale models. These Bayesian calibration-derived date ranges (HPD) are italicized to distinguish them from simply calibrated results.

The Intcal20 calibration curve [64] was used for terrestrial and freshwater invertebrate samples since this region falls within the Intertropical Convergence Zone (ITCZ) [following 65:1088]. The Marine20 curve [66] was used for the models including determinations from marine shell or human bone (Pain Haka, only). Marine20 is offset to Marine13 [67] by approximately 100 years. None of the ISEA determinations from marine shell or other marine-influenced samples (e.g., human bone) have associated ΔR values. The few published ΔR values for ISEA and neighboring portions of Australia and New Guinea document variability (calib.org/marine/) (e.g., Philippine values ranging from -215±50 for Janao Bay, Luzon Island, to -68±70 for Mindoro Straight). Therefore, we ran model iterations for sites with marine-influenced determinations with no correction value (ΔR 0±0); including the human bone-derived determinations from the Pain Haka, Flores, site (the authors provide isotopic data that is indicative of a marine component to diet). For Mussau, Petchey and Ulm's [68] ΔR value was adjusted based on Marine20 to -434±179 (95%). Ages obtained from animal bone are excluded if isotopic data have not been published or could not be obtained from the dating laboratory; these data are necessary for determining whether a mixed atmospheric-marine calibration is required.

Analysis was iterative. Single- or multi-phase calibration models [63] were created based on the number of radiocarbon determinations and published contexts. The models use Oxcal's Sequence, Phase, Boundary, Outlier_Model, Outlier, and Order commands (individual codes provided in SI). We used outlier commands in all of our models to down-weight the influence of potential outliers [see 69]. Two outlier commands were employed with the outlier distributions scaled between 0–200 years. The Charcoal Outlier command [63, 70] was applied to all radiocarbon determinations obtained from charcoal. The General Outlier command [63] was applied to determinations obtained from all other sample types. The convergence value for all models is ≥95%, indicative of stability [69:1043, 71]. To further assess the stability of the results, each model was run at least five times to ensure that consistent HPD were produced.

We used Oxcal's Order command to evaluate the chronological sequence of pottery introductions across the study area, and compare our results with the expectations of the OoT and WRM hypotheses. The start HPD were extracted from the individual site- and island-scale models and aggregated in new models with the Order command. The command provides pair-wise comparisons of the dates with probabilities that one date ($t_1$) is older than another ($t_2$). The chronological sequences presented (Figs 2 and 3) arrange dates so that one date has a probability >0.5 that it is older than all subsequent dates in the order.

## Supporting information

**S1 File.**
(DOCX)

## Acknowledgments

Steve Athens, Robert DiNapoli, Tom Dye, Derek Hamilton, Tony Krus, Peter Lape, and Christian Reepmeyer provided valuable feedback on this research. Seven anonymous reviewers also provided helpful comments at various stages. Chris Filimoehala created Fig 1.

## Author Contributions

**Conceptualization:** Ethan E. Cochrane.

**Data curation:** Ethan E. Cochrane, Timothy M. Rieth, Darby Filimoehala.

**Formal analysis:** Ethan E. Cochrane, Timothy M. Rieth, Darby Filimoehala.

**Investigation:** Ethan E. Cochrane, Timothy M. Rieth, Darby Filimoehala.

**Methodology:** Ethan E. Cochrane, Timothy M. Rieth.

**Project administration:** Ethan E. Cochrane.

**Supervision:** Ethan E. Cochrane.

**Writing – original draft:** Ethan E. Cochrane.

**Writing – review & editing:** Ethan E. Cochrane, Timothy M. Rieth.

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
