## [Decision Letter · Decision Letter 0]

19 Oct 2020

PONE-D-20-21080

Bayesian radiocarbon chronologies for initial pottery in Island Southeast Asia support multi-directional Neolithic dispersal

PLOS ONE

Dear Dr. Cochrane

Thank you for submitting your manuscript to PLOS ONE. After careful consideration, we feel that it has merit but does not fully meet PLOS ONE’s publication criteria as it currently stands. Therefore, we invite you to submit a revised version of the manuscript that addresses the points raised during the review process.

As is evident from the reviewers' comments, the manuscript requires a major revision.  The main criticism that has been raised by several reviewers that the critical re-evaluation of the available data is in the subjective treatment of certain aspects of the Bayesian models. There are also issues with some of the 'early dates' e.g. from Borneo, and the way by which certain radiocarbon dates were included while other were rejected (so the specific  'chronometric hygiene' criteria applied). 

Moreover, the above aspects also pertain to the interpretation and synthesis of the data, and the support/rejection of the relevant migration models. 

It is certainly not an easy task to handle it in the most objective manner possible, given the fact that the data quality is often poor. However, it is best in such cases to test several scenarios to pay close attention to the extent to which the results are affected by subjective decisions on the exclusion and inclusion of dates, and also whether certain models, such as 'Out of Taiwan' can be rejected. 

We look forward to receiving your revised manuscript.

Kind regards,

Ron Pinhasi

Academic Editor

PLOS ONE

Journal Requirements:

"Work by EEC was partially funded by The University of Auckland Faculty of Arts Performance Based Research Fund (no grant number or funder website). The funder had no role in study design, data collection and analysis, decision to publish, or preparation of the manuscript."

i) Please provide an amended statement that declares *all* the funding or sources of support (whether external or internal to your organization) received during this study, as detailed online in our guide for authors at http://journals.plos.org/plosone/s/submit-now.  Please also include the statement “There was no additional external funding received for this study.” in your updated Funding Statement.

ii) Please include your amended Funding Statement within your cover letter. We will change the online submission form on your behalf.

'The authors have declared that no competing interests exist. '. 

We note that one or more of the authors are employed by a commercial company: 'International Archaeological Research Institute Inc'.

4. We note that [Figure 1] in your submission contain [map/satellite] images which may be copyrighted. All PLOS content is published under the Creative Commons Attribution License (CC BY 4.0), which means that the manuscript, images, and Supporting Information files will be freely available online, and any third party is permitted to access, download, copy, distribute, and use these materials in any way, even commercially, with proper attribution. For these reasons, we cannot publish previously copyrighted maps or satellite images created using proprietary data, such as Google software (Google Maps, Street View, and Earth). For more information, see our copyright guidelines: http://journals.plos.org/plosone/s/licenses-and-copyright.

1.     You may seek permission from the original copyright holder of Figure [1] to publish the content specifically under the CC BY 4.0 license.  

Reviewers' comments:

Reviewer's Responses to Questions

**Comments to the Author**

1. Is the manuscript technically sound, and do the data support the conclusions?

Reviewer #1: Partly

Reviewer #2: No

Reviewer #3: Partly

Reviewer #4: Partly

2. Has the statistical analysis been performed appropriately and rigorously? 

Reviewer #1: Yes

Reviewer #2: No

Reviewer #3: No

Reviewer #4: Yes

3. Have the authors made all data underlying the findings in their manuscript fully available?

Reviewer #1: Yes

Reviewer #2: Yes

Reviewer #3: Yes

Reviewer #4: Yes

4. Is the manuscript presented in an intelligible fashion and written in standard English?

Reviewer #1: Yes

Reviewer #2: Yes

Reviewer #3: Yes

Reviewer #4: Yes

5. Review Comments to the Author

Reviewer #1: Overall, I thought this was a well-written and designed paper that provides some new insights into the oft-debated issue of Neolithic expansion in Island Southeast Asia, with implications for understanding the movement of groups into Near Oceania. I think this is a potentially publishable paper, though there are some (mostly minor) issues that the authors should consider prior to it being accepted for publication.

1. Abstract – says “with populations moving south into the islands” – but what “islands” are they referring to? They should specify here to give readers some context. Also, says “world that influenced human and environmental diversity from Madagascar to Easter Island”; true, but that’s only west-east; what about New Zealand and Hawaii (north-south)? Near bottom, should say “support” (not “supports”).

2. P. 4 – says that farming populations arrived in the Bismarck archipelago, Sulawesi, and Mariana Islands around 3450-3300 BP. Do the Marianas dates go back that far? What about Palau, which I believe has some coeval dates, though they would need to check the literature. In regards to this, it would be useful to have all major geographical locations referred to in the study also present in the map (e.g., Mariana Islands), especially for readers who may be unfamiliar with the region.

3. One issue for me is that the authors include potentially problematic dates such as residues from ceramics, freshwater shell, etc. They certainly don’t hide the fact they do so they’re not trying to be coy, and note that (in the case of residue dates) that their Bayesian analysis…(that include these dates) does not change their conclusions, although they would be, of course, less comprehensive.” They note too (p. 21) that they evaluate potentially problematic dates (e.g., unidentified charcoal, those with lack of localized correction values, etc.), but the parameters used to address these seem to focus primarily on whether there was a relatively secure stratigraphic context or that they were incorporated within the statistical models to help address various issues.

Much of the reasoning for including the problematic dates is that the sample numbers are low to begin with, at least in terms of having to examine such a geographically widespread series of events. I’m not saying that what they are doing is wrong per se, but it’s not overtly clear in the main text of the paper which of the dates were deemed more useful than others and what the comparisons might be between using the whole data set versus different variations of the analyses that use everything but pot residue dates, or everything but pot residues, freshwater shell, and unidentified charcoal, or….?

I guess my main concern is how the dataset might intrinsically change their interpretation of these population movements if certain dates (or sites that only have problematic dates) were removed from the analysis. This harks back to using a chronometric hygiene approach that Spriggs (1989) used to model Neolithic expansion in Island Southeast Asia more than 30 years ago (and that curiously, is not cited in the paper), and that Spriggs and others in the Pacific and elsewhere have used to examine the temporal efficacy of major migratory movements and colonization processes. If what they are doing is essentially chronometric hygiene, even if on a basic level, why don’t they simply call it what it is?

Overall, I would like to see some more effort on showing what would happen if certain problematic dates were excluded, and perhaps variations of these (e.g., what would the model look like if certain (or all) dates did not fit the criteria imposed?

Figure 1 – It would be helpful if this map also showed the general directions of the population movements they discuss, particularly since this is a major focus of the paper

Reviewer #2: This article is an attempt to use radiocarbon chronologies on Island Southeast Asian assemblages with the end result of supporting what they call a “Western Migration Route” out of mainland SE Asia as opposed to an expansion into Eastern island SE Asia which originated from Taiwan. By creating such a straw man hypothesis they believe their attempt at number crunching the dates will make a contribution to archaeological knowledge. Unfortunately, it does not. Neither of the authors is an authority of Asian Archaeology, and their ignorance of East Asian Archaeology comes through in what they attempted to do. Their approach is very clinical being void of assessment. It is more like a student exercise. The authors argue that an examination by archaeologists up until now have used “adoptive qualitative, and sometimes as hoc approach to interpreting radiocarbon determinations …where individual decisions to reject or accept particular dates are made by visually inspecting date distributions, often using inconsistently applied criteria, with the possible result of favouring preferred dispersal hypotheses”. Yet the reference given for this assertion has no bearing on who these errant archaeologists are that force dates to make their models look better. By taking 146 radiocarbon dates from 28 archaeological sites and using Bayesian model without any attempt at assessing the integrity of the assemblages they came from is a mistake. Matthew Spriggs pointed out over 30 years ago the need for what he termed “Chronometric Hygiene” – that is, assessing each date against a selected criteria including the lab it came from, material dated, sound archaeological knowledge of its context and the manner it was examined. That is what archaeologist do. To not do so is a mistake. By not doing so the old adage “Garbage in then garbage out” will be the result. To push all the dates into a machine and then come out and say the earliest occupation in the west without any detailed discussion of the archaeological record is reckless. Only a summary presentation of the record is presented in this manuscript. Having said that, even if the earliest Neolithic dates came from western island SE Asia, then how does that impact on an out of Taiwan model? On top of this there seems to also be mistakes in their dating. I note that they have dates for Lapita pottery from Emirau at 4088-3030 cal BP. I haven’t checked the article for this site but surely this is a mistake?

Since their exercise on dating there are new calibrations available, and also new Delta R must be assessed. This makes the radiocarbon calibrations here out of date.

Reviewer #3: This paper discussed the modelling of Bayesian radiocarbon chronologies for initial pottery in Island Southeast Asia. The authors argue that their analysis supports multi-directional Neolithic dispersal.

Although there is value to this work, I have reservations about the conclusions. The authors have included the term Bayesian in the title to inform the reader and imply a level of robusticity to their model building that belies the poor quality of the underlying data. Moreover, the paper title implies a much more definite conclusion about the directional nature of Neolithic dispersal than possible, even though the authors reiterate many times throughout the text that the information is flawed and that the numbers of dates are too few to make any firm conclusions. I would suggest modifying the title.

Specific comments

Lines 98-101: The authors state “…researchers have adopted a qualitative, and sometimes ad hoc, approach to interpreting radiocarbon determinations and associated pottery assemblages, where individual decisions to reject or accept particular dates are made by visually inspecting date distributions”. Similarly, lines 259-262 “…the Bayesian results generated from largely the same datasets suggest some of the qualitative analyses variably suffer from bias towards favoured hypotheses, spurious precision, and few clear evaluative criteria”. I am less convinced than the authors that their work doesn’t do the same as these earlier ‘chronometric hygiene’ methodologies, just with newer Bayesian modelling methods. For example,

• They repeatedly mention that freshwater shells are excluded because of uncertain reservoir offsets – yet, they provide no evidence of any site with a demonstrated freshwater offset. Freshwater and terrestrial shells have been used in many situations successfully (i.e., Higham and Higham 2009; Brook 2000). Tests on modern freshwater shells have also been undertaken by ANU at the site of Nagsabaran. These indicated minimal offset (Hung et al. 2011), and there is no firm reason to suspect that this is not so for the archaeological material.

• The authors also build models with unidentified wood charcoal with potentially 100’s of years of inbuilt age. The use of the charcoal outlier has limited value where additional constraints have not been placed on the date, such as stratigraphic sequence, short-lived dates from the same context, or other reliable date constraints. Four charcoal dates with inbuilt age constrained in a single-phase model using the charcoal outlier will always date too old.

• Some bone dates are excluded because of uncertain diet, while other samples have mixed marine corrections applied without justification for the values used (e.g. Wk-36556 – note this sample, and the other human bone dates from Pain Haka, don’t appear in Table S3 even though they are used in the OxCal code). In reality, few of the bone dates referenced by Cochrane et al. conform to rigorous standards of quality control. Where this information isn’t published, the authors should contact the labs concerned. It is unlikely that any bone dates measured in the last 20 years do not have some kind of quality assurance data associated with the results. Waikato dates do. Given many will refer to a synthesis paper, such as that written by Cochrane et al., the date information they report must be complete and correct.

• The authors are inconsistent with their treatment of residues. They consider residues to be too old for some sites but accept a younger residue date (Wk-14642) over an older residue date OZH-771 (lines 268-9).

Line 31, 116: The use of the term “Bayesian chronological analysis” is a broad description that gives the reader little idea about what Cochrane et al. have done. This paper discusses the highest probability distributions for start and end dates of ceramics in each site/region. The authors fail to mention, though they are aware of the sample number limitations, that this type of analysis is most effective when dealing with hundreds of dates of mixed material types (cf., Schmid et al. 2018), but can work with fewer dates. Sites with zero to four 14C dates, however, do not enhance the accuracy and precision of highest probability distributions and add little value to the discussion beyond that already provided by eyeballing the results. There are few sites presented in Table 1 with more than 4 dates, and typically, as is the case with many multi-date evaluations across the Pacific, these more extensive dating programs have been undertaken because of ongoing issues with the site chronology (e.g., Nagsabaran). This raises doubts about the integrity of the deposits, and even about our ability to correctly interpret some dates (e.g, residues, and freshwater shells – despite my comments above), both of which require additional investigation before they can be used in regional chronologies.

Bronk Ramsey (2009) similarly cautions against models that have small datasets;

"The more complex scaled models do, however, come with the risk of confounding effects, where 1 parameter is played off against another and this is particularly true if for small models. It is possible for the user to check for model misbehavior by looking at 3 aspects of the model output: the convergence (see Bronk Ramsey 1995), the posterior distribution for the scaling parameter u, and the posterior outlier probabilities for �i. A number of situations can arise:

• The convergence can be very slow. This is often associated with the scale of the offsets being hard to determine; in OxCal the model may never finish running at all if a satisfactory convergence is not achieved. In such circumstances, it may be necessary to use a simpler model.

• The distribution for u may be poorly constrained and extend right up to the upper limit. This is normally the consequence of using the scaled model for a data set that is too small to support it; the results will still provide a model average over the specified scales, but it would usually be better to use a simpler model in such circumstances".

The authors do not mention or evaluate any of the parameters mentioned by Bronk Ramsey. Instead, they mention (line 497) the agreement index (a parameter not used with outlier analysis) and make no mention of the convergence value (which should be >95%). Similarly, I tested some of their models and u was poorly constrained.

My suggestion, given the limited number of dates per site, is:

1. To remove the modelled date results/codes for each site and evaluate only on a island/region basis. This would require modification to Table 1.

2. Reduce the discussion to a regional evaluation, only mentioning individual sites when the dates are outliers (a definition of what the outlier values mean is also required , i.e., minor, major, or complete removal from the model outcomes).

3. The authors may find more value in a Bayesian approach that evaluates stratigraphy sequence and multiple phases. This should help constrain the dates. This will require collation of dates beyond that for first appearance of pottery.

Table 1: The start and end estimates are given at single year resolution. This is spurious accuracy given the nature of the dates involved. Suggest rounding to 10 years.

Throughout the text, when referring to their result compared to previous assessments, could the authors please add the result values in brackets to save the reader from having to refer to Table 1 continuously (e.g., lines 263-265, also 313-4).

I realise that the author (EC) has already commented about the dating of Nagsabaran. I do think the paper would benefit from more detail about the problems with this site. This is because Nagsabaran has the greatest number of dates available for comparison, and those responsible for the excavation of this site may be dismissive of informed interpretations that differ from their own.

The authors also mention biological, cultural and linguistic evidence that supports their model. Given the chronology by necessity relies on this additional data these should be elaborated on.

Lines 383-5: Statements should be modified to reflect confirmation of the age-estimates, not just to increase the precision, or outline that their interpretation is the most accurate (e.g, line 392).

Define terms – TAQ, TPQ, deltaR.

References

Brook, F., 2000. Prehistoric predation of the landsnail Placostylus ambagiosus Suter (Stylommatophora: Bulimulidae), and evidence for the timing of establishment of rats in northernmost New Zealand. Journal- Royal Society of New Zealand 30(3):227-241

HIGHAM, C. & T. HIGHAM. 2009. A new chronological framework for prehistoric Southeast Asia, based on a Bayesian model from Ban Non Wat. Antiquity 83: 125–44.

Hung H-C, Carson MT, Bellwood P, Campos FZ, Piper PJ, Dizon E, et al. The first settlement of Remote Oceania: the Philippines to the Marianas. Antiquity. 2011;85:909-26.

Schmid M, Dugmore A, Forest L, Newton A, Vésteinsson O, Wood R. How 14C dates on wood charcoal increase precision when dating colonization: The examples of Iceland and Polynesia. Quaternary Geochronology. 2018(48): 64–71.

Reviewer #4: The authors review chronostratigraphic sequences for 12 islands of ISEA and present a model built in a Bayesian framework in order to provide a better and more objective representation of a given chrono-cultural framework while expressly avoiding cherry-picking. Bayesian methods are now widely used by archaeologists but the authors provide the first formal review of the literature for dates of ceramics in ISEA. Well-dated contexts are crucial to understanding the timing and nature of the cultural shift to Neolithic in ISEA, and this phenomenon has to be characterized in terms of biological, linguistic and material evidence.

As such, this paper is an important contribution to the field and the prehistory of the region.

Assessing initial waves of migration is a difficult task, especially when dealing with complex chrono-cultural sequences, unclear associations between stratigraphic units and artefacts, and unidentified dated samples. The authors’ review and modelling of published dates is therefore valuable and will surely be used as a foundation to identify/discuss problems of cherry-picking or transparent report of contexts, and hopefully move forward with new dates. Data and code availability are exemplary here, and this makes it quite easy to reproduce the analysis on Oxcal.

The main downside of this critical re-evaluation of the available data is in the subjective treatment of certain aspects of the Bayesian models. In challenging the orthodox ‘Out of Taiwan’ model, the authors chose to validate the hypotheses of the ‘Western migration route’ (defined in lines 74 to 76) over the orthodox scenario. Unfortunately, I don’t think it is possible to favour a migration route over the other. In that sense, the data and models do not support the conclusion. A more accurate conclusion is that there is a high probability that both (out-of-MSEA) Borneo and (out-of-Taiwan) Itbayat have both been settled by people carrying/making ceramics before 4000 BP, but given the available data it is difficult to assess if these events are synchronic or not.

The authors use ‘early dates’ from Borneo to argue in favour of the WRM hypothesis, but I have reservations about the interpretations of the Bayesian models. Here are a few examples:

For MAD-1, the authors confuse the reader by including pre-ceramic dates in the date range distributions (but they don’t forget to remove the preceramic date from the Dalan Serkot cave in Luzon). Additionally, I strongly disagree with the authors that the boundary transition between layer 12 and ceramic layer 11a can be used to assess the appearance of ceramics in MAD-1: the gap between layer 12 (non-ceramic) and layer 11a (effectively associated with ceramics) is such that the boundary transition is not meaningful (8497-2725 BP at 2 sigma). The modelled date for this early ceramic layer 11a is 3058-2503 BP at 2s, and that is what should be used to assess the appearance of pottery on this site.

For Gua Sireh, the use of the start boundary to assess the beginning of ceramic period is also misleading. The two first dates in the sequence (CA725 and ANU7049) were performed a long time ago and are associated with CRA error > 220 years, which is reflected in the wide calibrated and modelled intervals. This artificially pulls back the start boundary of this sequence to 9678-3879 BP while the modelled dates within the single phase have the following HPD intervals: 4881-3637 and 4987-3644.

Following the migratory paths in the north and in the south, I would also argue that Luzon and Timor were first settled by ceramic makers around the same time, which also goes against the idea of an earlier southern migration route. Unfortunately, only two dates are available for Timor with the following HPD intervals: 4075-2809 and 3990-3590 at 2s, 3868-3268 and 3885-3690 at 1s. On the other hand, many dates are available for Luzon. Leaving aside potentially problematic ones, the earliest solid dates provided for Luzon are Wk-15648 (modelled date: 3920-2529BP at 2s, 3876-3436 at 1s), ANU-13016 (modelled date: 3885-2458BP at 2s, 3842-3390 at 1s), and NTU-3799 (modelled date: 3831-3580BP at 2s, 3822-3638 at 1s). I interpret the arrival of ceramic on both Luzon and Timor around the same time, between 3900 and 3600 BP.

The outlier model used to account for the difference of sample quality is appropriate. But because most samples are unidentified woods, the prior outlier probability of 1 (versus 0.05 for non-charcoal) dominates every model. Overall, this creates more uncertainty when assigning dates for each region, which adds to the difficulty of interpreting the results in terms of regional migratory patterns. This is particularly true for the samples from Borneo, Java and Sulawesi, with 93.9 % of the dates obtained from non-identified charcoals and therefore evaluated with the maximum outlier probability (only 67.3% in Batanes and Luzon; 63.2% for Timor, Flores, and Maluku; 52.2% for Mussau).

Beyond the debate between two restricted hypotheses of WMR and OoT, I strongly agree with the interpretation that the patterns of initial migrations associated with ceramics in ISEA are multi-directional. I think the discussion and conclusion should actually be bolder in emphasizing this aspect, and they could go beyond the two main models to assess the complexity of Neolithic dispersals in terms of people, languages, and material culture. As a consequence, I also suggest that the authors address more directly how their results fit with language history (1) and genetic history (2).

1- The linguistic tree of the Austronesian family published by Gray et al in 2009 shows little deep structure in ISEA, suggesting either a very rapid expansion or a series of multiple expansions. Because the chronological framework presented here correspond to spread of Neolithic populations in ISEA over more than 1 millennium, this could also be used to privilege the hypothesis of multiple expansions. Nonetheless, linguistic trees (based on classical comparative approaches or on Bayesian phylogenetic methods) generally show that Taiwanese languages is the first branch, and that contradicts the hypothesis of a primary southern China route.

2- The authors should address more directly how their results fit with aDNA data. Lipson et al 2014 argued in favour of the OoT model, but Soares et al. 2016 have challenged their conclusion on the basis of limited source material and recent admixture time for the Taiwan ancestry in ISEA, but both Soares et al and Brandão et al 2016 emphasize that both expansions, through Taiwan and MSEA, were due to small-scale migrations and do not privilege one hypothesis over the other.

Other comments:

- The manuscript is intelligible and written in standard English. As a non-native English speaker, it is hard to assess the quality of writing.

- The name of islands and islands group should be consistent across the manuscript and SI (example: Molucca Islands / Maluku, etc.)

- As stated above, I advise that the HPD for Um Kapat Papo / Gebe Island be removed from Fig 2 and 3 as these dates “were excluded from the analysis” (line 1913 of Supp Info).

- Table S1 and S2 are useful but it is unclear which dates were used in the statistics. It would be valuable to have more details on this probability test and the underlying data.

- In table 1, change the number of samples dated from ECB (Mussau): n=3 instead of 23.

6. PLOS authors have the option to publish the peer review history of their article (what does this mean?). If published, this will include your full peer review and any attached files.

Reviewer #1: No

Reviewer #2: No

Reviewer #3: No

Reviewer #4: No

---

## [Author Response · Author response to Decision Letter 0]

23 Jan 2021

Almost all reviewer comments have been accepted and addressed. Full details are in the Response to Reviewers document.

---

## [Decision Letter · Decision Letter 1]

18 Feb 2021

PONE-D-20-21080R1

Bayesian radiocarbon chronologies for initial pottery in Island Southeast Asia support multi-directional Neolithic dispersal

PLOS ONE

Dear Dr.  Cochrane,

Thank you for submitting your manuscript to PLOS ONE. As indicated in the attached comments, two of the reviewers provide additional comments and suggested minor edits.

Please submit your revised manuscript by the 31st of  March.  If you will need more time than this to complete your revisions, please reply to this message or contact the journal office at plosone@plos.org. Please include the following items when submitting your revised manuscript:

We look forward to receiving your revised manuscript.

Kind regards,

Ron Pinhasi

Academic Editor

PLOS ONE

Reviewers' comments:

Reviewer's Responses to Questions

**Comments to the Author**

1. If the authors have adequately addressed your comments raised in a previous round of review and you feel that this manuscript is now acceptable for publication, you may indicate that here to bypass the “Comments to the Author” section, enter your conflict of interest statement in the “Confidential to Editor” section, and submit your "Accept" recommendation.

Reviewer #1: All comments have been addressed

Reviewer #3: (No Response)

Reviewer #4: All comments have been addressed

2. Is the manuscript technically sound, and do the data support the conclusions?

Reviewer #1: Yes

Reviewer #3: Partly

Reviewer #4: Yes

3. Has the statistical analysis been performed appropriately and rigorously? 

Reviewer #1: Yes

Reviewer #3: Yes

Reviewer #4: Yes

4. Have the authors made all data underlying the findings in their manuscript fully available?

Reviewer #1: Yes

Reviewer #3: Yes

Reviewer #4: Yes

5. Is the manuscript presented in an intelligible fashion and written in standard English?

Reviewer #1: Yes

Reviewer #3: Yes

Reviewer #4: Yes

6. Review Comments to the Author

Reviewer #1: I think the authors have done a really nice job in responding to my and all of the other reviewers’ comments. I’m satisfied with the current draft pending a few other minor revisions.

Keywords: these should not replicate what is seen in the paper title since they are both queried separately and will allow the paper to gain greater exposure when people search for different terms.

Figure 1 caption – “Esri” should be capitalized, no? And maybe I’m missing something, but I don’t see this Figure anywhere in the document. It’s not embedded in the generated PDF that they submitted nor as a link to other material.

While I cannot assess the map, it should make sure to include the directional arrows that were requested earlier and might be also helpful to show date ranges that were gleaned from their study which would be helpful to readers.

There seems to be extra spaces added after most colons throughout manuscript

Line 180 – extra space after “assemblage”

Line 192 – extra period at end of sentence

Reviewer #3: I welcome the comments in this paper that force a reconsideration of supposed orthodox views of Neolithic dispersal through ISEA. The authors have demonstrated clearly the inherent biases in the majority of previous chronological evaluations and, therefore a need to reconsider the evidence and reinvestigate the regions chronology with an open mind.

Lines 315-317: I absolutely agree with the statement “Comparing these chronologies with the Bayesian results generated from largely the same datasets suggests some of the qualitative analyses variably suffer from bias towards favored hypotheses, spurious precision, and few clear evaluative criteria”.

I especially welcome the assessment of Nagsabaran using this quantitative method (see comment on line 399-401) and the authors provide a useful comparison with the Mussau date set.

This paper appears to negate both the OoT and the WRM hypotheses and opens up the ability to have a new discussion about the chronology. As I outlined in my previous review, I think this is important to highlight in the title, more so than the Bayesian analysis aspect which is overused as a catch phrase which gives the reader little information about what makes this paper special. Suggestion: “The first quantitative assessment of radiocarbon chronologies for ISA support multi-directional Neolithic dispersal”.

General comments about revised document:

Replace the term “estimate” throughout both the main text and the supplementary. Estimate definition = “roughly calculate or judge the value, number, quantity, or extent of”. A radiocarbon date is not an estimate, it is a measured value with an error term, e.g., line 294…instead of estimate “The initial appearance of ceramics at PA1 on Pulau Ay dates to between 3740-3020 cal BP (95.4%).

Similarly, a certain level of negativity about the dates beyond fact creeps into the text. For example, the use of “sometimes” in Line 237. Suggest instead “dates to between 5440-4270”. Suggest careful evaluation of comments about age ranges.

Table 1. Highlight the regional/Island results

Note: in table the number of samples from Luzon is 24, but Nagsabaran is 27. The reason for this is discussed in the text, but a footnote to the table would clarify this to the reader.

Line 175: The residue-derived determinations do not add 250 years to the dates. The range shifts by ~250years but the new range overlaps include part of the old range.

Line 175: delta13C should be superscript (throughout text and tables).

Throughout the paper: The use of a deltaR of 0 with Marine20 is problematic. Marine20 is offset relative to Marine13 (Reimer et al. 2013) by at least 100 years (Heaton et al. 2020), so a correction of at least this is required to keep the calibrations compatible with evaluations that have previously used Marine13. Evidence gathered from across the South Pacific suggests a minimum -150yr offset between the old and new curves for the late Holocene. Modern shell values (Calib.org) indicate an average of -106+/-50 for as an average for the ISWP region. If the authors are concerned about a temporal variability in the deltaR value as suggested by Petchey (2019), one option would be to use a variable deltaR command in OxCal, eg.

Curve("ShCal13","ShCal13.14c");

Curve("Marine13","Marine13.14c");

Delta_R("Local offset",U(-400,400));

Supplementary files

Lines 3463 onwards: The model for Lena Hara uses Marine13 and a deltaR of 0. This is not compatible with the use of Marine20 elsewhere (see comment above). Also, Um Kaput Papo, Uatlandi and ECB.

SI figures: Colour the regional plots a different shade or colour from the site distributions.

S2 figure: Specify as footnote which model result for Nagsabaran has been used.

S3 Table:

delta13C should be superscript

Lab codes need to be standardized, e.g., ANU-001 or ANU001, not both

Bone material should be specified, i.e., dentine vs enamel, gelatin vs ultrafiltered.

Stable isotope data for the bone dates should be included (they are discussed, but I cannot find the specific data anywhere in the table. These have an impact on the reliability of the result. References to bone interpretation should be given.

Molaccus – estuarine spelt wrong.

Specify in Table S3 which deltaR values have been used. The references given in the table for the deltaR values used were all calculated pre-Marine20, therefore an updated value using the base data from those references should be given to ensure clarity.

Line 2153: Please define what poor convergence means. You cannot assume that the reader is familiar with these definitions.

Reviewer #4: I am pleased to see that most of my comments were taken into account. In particular the authors have 1) elaborated on linguistic and genetic literature using general comments and references that I provided, and 2) have highlighted the need for better dates. Most importantly, the authors have agreed to rerun their models and to change their conclusion, which better fits the data at hand.

7. PLOS authors have the option to publish the peer review history of their article (what does this mean?). If published, this will include your full peer review and any attached files.

Reviewer #1: No

Reviewer #3: No

Reviewer #4: No

---

## [Author Response · Author response to Decision Letter 1]

21 Mar 2021

All responses to reviewers are included in the Response to reviewer document uploaded.

---

## [Decision Letter · Decision Letter 2]

27 Apr 2021

The first quantitative assessment of radiocarbon chronologies for initial pottery in Island Southeast Asia supports multi-directional Neolithic dispersal

PONE-D-20-21080R2

Dear Dr. Cohrane,

We’re pleased to inform you that your manuscript has been judged scientifically suitable for publication and will be formally accepted for publication once it meets all outstanding technical requirements.

Kind regards,

Ron Pinhasi

Academic Editor

PLOS ONE

Additional Editor Comments (optional):

Reviewers' comments:

Reviewer's Responses to Questions

**Comments to the Author**

1. If the authors have adequately addressed your comments raised in a previous round of review and you feel that this manuscript is now acceptable for publication, you may indicate that here to bypass the “Comments to the Author” section, enter your conflict of interest statement in the “Confidential to Editor” section, and submit your "Accept" recommendation.

Reviewer #1: All comments have been addressed

2. Is the manuscript technically sound, and do the data support the conclusions?

Reviewer #1: Yes

3. Has the statistical analysis been performed appropriately and rigorously? 

Reviewer #1: Yes

4. Have the authors made all data underlying the findings in their manuscript fully available?

Reviewer #1: Yes

5. Is the manuscript presented in an intelligible fashion and written in standard English?

Reviewer #1: Yes

6. Review Comments to the Author

Reviewer #1: All of my previous comments have been addressed. Note that there is a space needed between “datesfor” on Line 495

7. PLOS authors have the option to publish the peer review history of their article (what does this mean?). If published, this will include your full peer review and any attached files.

Reviewer #1: No

---

## [Editor Report · Acceptance letter]

10 May 2021

PONE-D-20-21080R2 

The first quantitative assessment of radiocarbon chronologies for initial pottery in Island Southeast Asia supports multi-directional Neolithic dispersal 

Dear Dr. Cochrane:

I'm pleased to inform you that your manuscript has been deemed suitable for publication in PLOS ONE. Congratulations! Your manuscript is now with our production department. 

Kind regards, 

on behalf of

Dr. Ron Pinhasi 

Academic Editor

PLOS ONE